# In Situ Observation and Phase-Field Simulation Framework of Duplex Stainless-Steel Slab during Solidification

**DOI:** 10.3390/ma15165517

**Published:** 2022-08-11

**Authors:** Tong Wang, David Wexler, Liangliang Guo, Yangfan Wang, Huijun Li

**Affiliations:** 1School of Mechanical, Materials, Mechatronics and Biomedical Engineering, University of Wollongong, Northfields Ave., Wollongong, NSW 2522, Australia; 2BaoWu Steel Group, Baoshan, Shanghai 201900, China

**Keywords:** high-temperature confocal microscope, phase-field simulation, duplex stainless steel, cooling rate

## Abstract

The melting and solidification process of S32101 duplex stainless steel (DSS) was investigated using high-temperature confocal microscopy (HTCM). The method of concentric HTCM was employed to study microstructure evolution during the solidification process of S32101 DSS. This method could artificially create a meniscus-shaped solid–liquid interface, which dramatically improved the quality of in situ observations. During the heating stage, γ-austenite transformed to δ-ferrite, and this transformation manifested itself in the form of grain boundaries (GBs) moving. The effects of cooling rate on the solidification pattern and microstructure were revealed in the present research. An enhanced cooling rate led to a finer microstructure, and the solidification pattern changed from cellular to dendritic growth. As the temperature decreased, the commencement and growth of precipitates were observed. In this paper, the experimental data, including parameters such as temperature, cooling rate, and growth mode, were used as the benchmark for the simulation. A simulation framework using Micress linked to a 1D heat transfer model enabling consistent analysis of solidification dynamics in DSS across the whole cast slab was established. Simulating the dendrite growth and elemental segregation of DSS at specific cooling rates shows that this framework can be a powerful tool for solving practical production problems.

## 1. Introduction

S32101 (US designation) is a duplex stainless steel (DSS) with low content of Ni and high content of N. High contents of Mn and N were added to replace expansive Ni to reduce the cost and improve mechanical properties and corrosion resistance. It is widely used in the chemical industry or offshore technologies, where a combination of high corrosion resistance and good tensile strength is required [1,2].

Many factors, such as cooling rate, can affect the microstructure of steel during the solidification process. Importantly, the microstructure can affect the mechanical properties of S32101 DSS and finally influence the quality of the slab. The dual-phase structure is the root of the excellent properties of DSS. Mazumdar and Kary [3], from various perspectives, such as heat transfer during crystallization and solid–liquid interface tribology, expounded on the control of the continuous casting process and pointed out that controlling the solidification conditions controlled the ratio of δ and γ to obtain products with excellent quality.

High-temperature confocal microscopy (HTCM) was used to observe the solidification process of S32101 DSS in the present research, and this technique can provide a so-called in situ observation. Zhao and Sun [4] applied laser scanning confocal microscopy (LSCM) in their research on DSS. They observed and discussed the effect of different N contents on the solid-state δ 
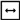
 γ phase transformations and concluded that N could hinder the migration of δ/γ interface boundaries (IBs).

Computational simulation can also be used predict the solidification process from experimental observation. The Microstructure Evolution Simulation Software (Micress^®^, Version 7.1) is a software package that calculates the growth of newly formed phases during solidification or phase transformation of a metallurgical system on a temporal or spatial scale. The software is based on the multiphase-field concept, developed by ACCESS e.V. (affiliates of Aachen University, Aachen, Germany) scientists since 1995. It is widely used in solidification, solid-state transformations, grain growth, recrystallization, heat treatment, etc. [5,6]. The solidification process and subsequent solid-state phase transformation always start with nucleation, which is the first step of forming a new phase. This process is dominated by thermodynamic driving forces, diffusion, and interfacial curvature. The backbone of Micress is the multiphase field method for multicomponent alloys, which enables the treatment of multiphase, multigrain, and multicomponent problems. As a well-established commercial phase-field simulation software, Micress provides a simulation platform for materials researchers without a computer science background.

In continuous casting, liquid steel is poured into a water-cooled copper mold, with the high heat extraction rate chilling a solid steel shell that contains the yet-to-be solidified liquid in the center. This shell is continuously extracted from the copper mold, the rate of which is an important operating parameter. Secondary cooling through water mist is applied to the shell as it exits the mold, thus inducing sufficient cooling to ensure complete solidification. The solidification process occurs with a liquid-to-solid phase transformation, and segregation of alloying elements such as Cr and Ni, as well as N and Mo for DSS in particular, could lead to defects. For the continuous casting of S32101 DSS, a deeper understanding of the effect of thermodynamic and kinetic parameters on the rate and mechanism is the key to improving the final product quality.

Many studies have been published about DSS with respect to the influence of the different alloying element contents. Scarce research has been conducted to investigate the thermodynamic process from melting to solidification. This paper aimed to reveal the melting process of S32101 DSS and the solidification process regarding different cooling rates through ‘in situ’ observation; furthermore, on the basis of the experiments, a phase-field simulation framework was established to determine whether concentric solidification can be used to provide a basis for benchmarking the relationship between cooling rate and solidification structure and to what extent this can be used to provide data for developing the Micress phase-field simulation framework.

## 2. Experimental and Modeling Approaches

### 2.1. Experimental Approaches

As an important tool widely used to observe solidification and phase transformation, HTCM sometimes reduces the observation surface because the sample forms semicircular droplets in the crucible. If there is further obstruction by oxides or precipitates, the availability of observational data will be significantly reduced. The design of a concentric solidification method can tackle this problem.

#### 2.1.1. Concentric HTCM

In 1961, Minski [7] established the principles of LSCM, which are widely applied in biological sciences. This equipment has been applied to investigate many fields of research, including the morphology of solidification and an analysis of the δ/γ interphase boundary dynamic behavior in low-carbon steels [8,9], inclusion agglomeration and engulfment in steel melt/solid interface [9,10], and crystallization of oxide melts [11]. Since the concentric HTCM was designed, significant developments in peritectic reaction and transformation have been achieved. Stephan [12,13] employed the concentric HTCM method to investigate the peritectic transformation of Fe–C steel and cast new light on the reason for massive-type phase transformation with thermodynamic and kinetic arguments. Lasertec Corporation manufactures the HTCM equipment in Yokohama, Japan. A 1.0 kW halogen lamp is located at a focal point in a gold-plated ellipsoid cavity in the lower half of the infrared furnace. The sample was heated by radiation in this chamber. A gold-coated ellipsoidal chamber concentrated the heat onto the surface of the sample located at the other focal point of the furnace chamber. Figure 1 shows the components of the 1LM21H type HTCM with an oxygen detector, monitors, and proportional–integral–derivative (PID) controller.

Reid et al. [14,15] developed and described the concentric solidification technique to improve the quality of in situ observation of HTCM. A thin sample is partially melted in the center, forming a pool of melt surrounded by a solid rim. A platinum sample holder holds the crucible in place, and a type B thermocouple is attached to the crucible. Since the in situ observation of HTCM is limited to the sample surface, the thickness becomes an important parameter. After several attempts with different sample thicknesses, ~250 µm was determined to be the minimum thickness for concentric solidification experiments on S32101 DSS. On the one hand, if the thickness is less than 250 μm, the liquid pool is easily ruptured. On the other hand, if the thickness exceeds 300 µm, the output power required to melt the sample will be too high to be achieved.

In DSS, initial melting is uneven due to segregation during the casting of the slab, which was not previously homogenized. Care needs to be taken for the rate of further heating to achieve a uniform liquid pool in the central region of the sample. With care, a pool of diameter around 5 mm can be achieved with a planar interface between liquid and solid. This then becomes the starting condition for all subsequent imposed cooling rates. The range of potential solidification microstructures possible includes planar at low cooling rates, and then cellular as the cooling rate increases. The subsequent transition is a dendritic microstructure which reverts to cellular and planar if a sufficiently high cooling rate is achieved. In the concentric solidification technique, the maximum controlled cooling rate that can be achieved is 500 to 1000 °C/min, far below the rate required for reverting to cellular/planar structures.

The sample and the holder were inserted into the top half of the furnace chamber, and high-purity argon gas (99.9999 vol.%) was flowed through the chamber. The gas was passed through a stainless-steel tube filled with titanium turnings to ensure a high-integrity inert gas, held at a temperature of 850 °C, before entering the furnace chamber. The oxygen level was reduced to 10–14 ppm before running the experiments. Moisture was also removed when the gas flowed through the Ti furnace.

The Omron ES100P digital PID controller panel was used to set the program profile before the experiments, as well as change the temperature during the experiments. The temperature program is shown in Figure 2a. All HTCM experiments were based on this procedure. The whole procedure can be divided into five stages. The first stage is to stabilize the equipment. The second stage is the heating stage, which involves heating to 1300 °C at a rate of 150 °C/min. The target temperature is below the melting point; one reason is to avoid overshooting which could leading to sample penetration, and another reason is to reserve enough operating range for the third step. Stage 3 is the most important because the liquid pool is formed at this stage, and it is necessary to artificially slowly adjust the temperature until a stable liquid pool of sufficient size is formed. The fourth stage is the heat preservation stage to form a stable solid–liquid interface. The last stage is the cooling stage with five different cooling rates, 5 °C/min, 10 °C/min, 50 °C/min, 80 °C/min, and 150 °C/min, to investigate the effect of cooling rate on solidification microstructures.

#### 2.1.2. Materials

The samples used in present research were provided by Baosteel (as shown in Figure 2b), named 1.4162 (X2CrMnNiN21-5-1) according to the EN standard. This grade of steel is widely used in extreme environments which require both excellent mechanical properties and high resistance to intergranular and pitting corrosion. The chemical composition of S32101 DSS is given in Table 1. All the samples were cut into slices. The diameter of a single slice was 9.7 mm, and the thickness was 0.25 mm. To prepare samples, a TechCut 5TM precision high-speed saw with water cooling was used to cut the sample into specific thicknesses before grinding on 800, 1200, 2400, and 4000 SiC papers. The samples were polished with standard metallographic techniques using a 6 μm diamond paste and 1 μm diamond paste. A fine sample surface can provide better images, especially in the heating stage or for solid-state studies.

### 2.2. Modeling Approaches

#### 2.2.1. Thermodynamics: Thermo-Calc

Released over 30 years ago, Thermo-Calc has become one of the most popular thermodynamic calculation software packages in the world. Thermo-Calc simulations can improve design during manufacturing and aid in selecting heat treatment temperatures. Calculations can predict experimental outcomes and minimize the number of experiments. In this study, the 2021b version of Thermo-Calc with the TCFE9 database was used for the calculation of phasors and their compositions, transition temperatures, and phase diagrams.

Another use of Thermo-Calc was the generation of optimized thermodynamic and kinetic datasets encapsulated in a GES5 file. This was applied in the subsequent phase-field simulation as a basis for microstructure development.

#### 2.2.2. Phase-Field Simulation: Micress

At the interfaces, the phase-field variables change continuously over an interface thickness *η*, which could be significantly large compared with the atomic interface thickness and tiny compared to the microstructure length scale. Their time evolution is calculated by a set of phase-field equations derived from the minimization of the free energy functional [16].
(1)$${\dot{\emptyset }}_{\alpha }=\overset{v}{\underset{\beta \neq \alpha }{\sum }}M_{\alpha \beta }\left[\frac{\sigma_{\alpha \beta }}{v}\left(\frac{1}{2}\left(\nabla^{2}\emptyset_{\alpha }-\nabla^{2}\emptyset_{\beta }\right)+\frac{\pi^{2}}{2\eta_{\alpha \beta }^{2}}\left(\emptyset_{\alpha }-\emptyset_{\beta }\right)\right)+\overset{n}{\underset{\gamma \neq \alpha \neq \beta }{\sum }}J_{\alpha \beta \gamma }+\left|\nabla \emptyset \right|{∆}_{\alpha \beta }\right],$$
(2)$$J_{\alpha \beta \gamma }=\frac{1}{2}\;\left(\sigma_{\beta \gamma }-\sigma_{\alpha \gamma }\right)\left(\frac{\pi^{2}}{\eta^{2}}\emptyset_{\gamma }+\nabla^{2}\emptyset_{\gamma }\right),$$
(3)$$M_{\alpha \beta }={\tilde{M}}_{\alpha \beta }\frac{8\eta }{\pi^{2}},\;$$
where $J_{\alpha \beta \gamma }$ denotes the higher-order terms accounting for the interaction with additional phases in triple- or multiphase regions.

For the sake of simplicity, only the isotropic formulation is shown here. In Equation (3), $M_{\alpha \beta }$ is the mobility of the interface as a function of the interface orientation, $K_{\alpha \beta }$ is related to the local curvature of the interface, and $\sigma_{\alpha \beta }^{*}$ is the anisotropic surface stiffness. The thermodynamic driving force and the solute partitioning are calculated separately using the quasi-equilibrium approach [16] and are introduced into the equation for the multiphase fields (Equation (3)). If concentration coupling is activated, an explicit 1D temperature field in the *z*-direction can be optionally defined by the keyword ‘1d_temp’. With the 1D temperature field, heat flow and latent heat release are solved explicitly. For numerical reasons, the temperature in the 1D temperature field is calculated using a direct, explicit solver with a default kinetic coefficient [17]. The default value is preconfigured for typical casting processes and needs not to be changed in most cases.

The temperature field can be evaluated in a linear (cartesian), cylindrical (cylindrical), or polar (polar) coordinate system; however, in the present research, a linear setup was used. Furthermore, we ran simulations at 0 mm, 10 mm, and 50 mm in the cast slab.

As shown in Table 2, the thermodynamic data were provided by online coupling to the database TCFE9 using the elements given in Table 1 and the phases in Table 2. Similarly, diffusion data for *γ* and *δ*, including cross-terms, were taken from the MOBFE4 mobility database.

## 3. Results and Discussion

DSSs have a so-called dual-phase structure, which, for S32101, contains about 50% volume face-centered cubic (FCC) γ islands and 50% volume body-centered cubic (BCC) δ matrix. After the samples were ground and finally polished, the surface of S32101 DSSs was observed with an optical microscope using a 0.04 µm particle size silica gel cloth. For the S32101 DSS, as shown in Figure 3, even without etching, the dual-phase structure was prominent, with almost equal proportions for each phase.

A property diagram, which reveals the response of all phases to temperature change, was calculated using Thermo-Calc under equilibrium conditions in the present research. The phase equilibria were calculated with a temperature range from 500 °C to 1500 °C. As shown in Figure 4, as temperature decreased, the liquid phase first solidified to δ, and, after solidification was finished, δ remained for a while and then partially transformed to γ. As temperature decreased, under equilibrium solidification conditions, the precipitation of secondary phases, such as chromium nitrides, σ phase, and χ phase, proceeded.

According to the thermodynamic calculation of S32101 DSS, the δ-to-γ phase transformation occurred around 1270 °C. As shown in the figure, the gas phase formed after the steel fully solidified and before the commencement of δ-to-γ solid-state phase transformation, approximately in the range of 1280–1430 °C. For S32101 DSS, nitrogen is used as an alloying element, and its content is very high. The solubility of nitrogen was the most significant factor in the gas pore formation. The solubility of nitrogen in the solid phase is much smaller than in liquid. During the solidification process, the nitrogen was rejected from the liquid phase and formed nitrogen gas pores. The nitrogen gas pore formation was also influenced by nitrogen content, cooling rate, partial pressure, etc. Hence, the temperature range of gas-phase formation is not fixed.

### 3.1. Concentric HTCM Experimental Results

The HTCM results are discussed according to the timeframe from heating to cooling, and the time is counted from the start of the experiment. The temperature values shown in the concentric HTCM images are those of the sample edges. Since the diameter of the sample was 9.7 mm, the temperature of the center was higher than that of the edge.

#### 3.1.1. Heating Stage

In Figure 5, a red circular area P is used as a reference marker. As shown in frames (a) and (b), the color of δ darkened, making it straightforward to distinguish γ and δ in this period. Frame (c) shows that the morphology of γ changed with the temperature rising. The γ and δ interface boundaries were apparent. The γ → δ phase transformation was not clear during the heating stage except for the morphology change. Frames (d) and (e) show the growth of δ/δ grain boundaries (GBs). The schematic diagram of frame (f) shows the morphology of γ. The γ accumulated along the grain boundaries of δ with a finger- or needle-like pattern. The intragranular γ-cells maintained an island pattern in the δ matrix. The grain boundary motion was in the form of a γ → δ phase transformation.

As the temperature increased, γ fully transformed to δ, and δ GBs showed up. In Figure 6, the red dotted line represents the δ/δ GBs, while the white dotted line represents the solid/liquid IBs. In frame (b), a new δ GB showed up; after the growth of new δ GBs, the sample started melting, as shown in frame (c). The liquid phase in frame (d) was located along the δ GBs.

As the temperature increased, γ transformed to δ. The GBs enlarged and slipped before melting, and thermal etching appeared, as shown in Figure 7. These phenomena indicated that the steel would melt soon.

When polished samples are heated to high temperatures and exposed to an inert atmosphere (e.g., argon gas in present experiments), the GBs will slip. Figure 7 shows the thermal etching in the solid rim of concentric experiments. As indicated in Figure 7b, grooves formed at elevated temperatures and remained intact after cooling. Figure 7 also reveals mobile grain boundaries and stationary grain boundaries. In frame (a), the GBs slipped in one direction and later toward the opposite direction (frame (b)). Moreover, the stationary δ/δ GBs enlarged, which is called thermal grooving. The slipping of GBs is called thermal etching, leading to the formation of a new δ-cell, as shown in frame (c). After the temperature reached the melting point, the liquid phase firstly showed up inside the stationary δ GB. Frames (e) and (f) show the commencement and growth of the liquid phase. Areas I, II, and III represent the liquid phase in different places. I indicates the liquid phase inside GBs, II indicates the liquid within two δ GBs, and III is the inside of a δ-cell. Later, all the liquid phases merged into a liquid pool. The concentric solidification experiment improved the quality of in situ observations due to the minimization of the meniscus and the increase in the length of the visual interface between liquid and solid [15].

#### 3.1.2. Solidification Process

In the continuous casting process, the quality of products is highly related to the grain structure during solidification, including grain size, morphology, and orientation [18]. The cooling rate can influence the solidification behavior and the solid shapes during the solidification process.

As shown in Figure 8, with a cooling rate of 5 °C/min, the sample solidified initially with a planar solidification front due to the required time for solute gradients to form as the solid grew and for instability to commence. After this initial planar solidification, the structure destabilized into a cellular solidification structure. As shown in frame (a), the sample started by solidifying from outside the liquid pool to inside gradually, and the solid/liquid interface boundary (S/L IB) moved slowly. The distance of S/L IB motion (D) under 5 °C/min was around 588 μm. Frame (b) shows that the δ solid grew with a columnar structure. The GBs were easy to observe, and the grain size was large. As shown in frame (c), the cellular structure could be observed as dendrites without secondary arms. Because of the prolonged cooling rate, the growth of primary arms was even toward the center of the liquid pool, with no space for secondary arms to grow. As shown in frame (d), the cellular structures were compact, and the content of alloy elements and gas precipitates was low; hence, gas pores were unlikely to form in this area. The temperature gradients in the liquid promoted the growth of cellular grains. The cooling rate also played an essential role in the growth of a columnar structure. As the cooling rate increased, the length of the columnar structure decreased.

Primary dendritic arms grew beneath the liquid surface, but secondary dendritic arms (SDAs) growing perpendicular to the growth direction appeared as a series of aligned islands at the liquid surface. When the cooling rate increased to 10 °C/min after the initial periods of planar and cellular solidification, there was a transition to dendritic solidification, as shown in frames (a–c). Solidification started in the S/L IB. As shown in frame (a), the D of S/L IB was 380.25 μm; right after the growth of the columnar δ, the secondary dendritic arms of δ commenced. In frames (b) and (c), the growth of primary and secondary dendritic arms can be observed. The appearance of SDAs followed the growth of a primary dendrite, and the growth of secondary arms was orientated. The arrows in frame (b) point to the preferred growth direction of primary and secondary arms. The microstructure under a 10 ℃/min cooling rate showed large graininess and dendritic shapes.

For the sample cooled at 50 °C/min, as shown in Figure 9d, the D was 87.75 μm. As shown in frame (e), the dendrites grew in the center of the liquid pool and were surrounded by several δ GBs, which can be observed clearly in frame (f). Under the cooling rate of 50 °C/min, the grain size of dendritic and cellular δ was still significant.

For the sample cooled at 80 °C/min, as shown in Figure 9g–i, solidification started in the S/L IB, and SDAs emerged from outside the liquid pool to the inside within 10 s. The microstructure comprised cellular structure domains with a small number of needle shape particles and no more dendritic microstructure. In frame (g), the D was 93.6 μm; right after the commencement of dendritic δ, the δ solids grew along the S/L IB. In frame (h), within the liquid pool of the sample, the solidification proceeded and then grew into a long and slender needle-like or dendritic pattern (frame (i)).

For the sample with a 150 °C/min cooling rate, as shown in Figure 9j–l, the region of cellular solidification was no longer observed. Instead, a small band of planar solidification destabilized directly to dendritic solidification with a smaller primary and secondary arm spacing, and solidification first started along the S/L IB. The solids simultaneously emerged both at the center of the liquid pool and at the edge of the S/L IB. The morphology comprised grains of finer cellular shape without evidence of a dendritic microstructure. The grain size became finer and smaller than outside of the liquid pool. In frame (a), the D was 76.05 μm, and the columnar δ stopped growing shortly after. The nucleation started inside the liquid pool and along the S/L IB shortly after. In frame (k), a number of secondary dendritic arms commenced inside the center liquid pool, whereas, in frame (l), all solid δ showed up with a cellular structure.

These results allowed studying the characteristics of the solidification microstructure response to thermal conditions. They demonstrated that concentric solidification offers a good platform for studying duplex stainless steels, forming a good testbed for subsequent simulations.

### 3.2. Micress Simulation

The most significant development of the present research was establishing a framework to utilize the combination of experiments and simulation to study the solidification process of S32101 DSS. This simulation framework can reduce the experimental workload dramatically. Concentric HTCM was also used to authenticate the accuracy of the simulation.

Micress simulations followed the same conventions, with the bottom of the domain oriented in the direction of the slab surface and the top of the domain pointing to the center of the slab. Figure 10 shows the selected outputs from the region of the chill surface where (a) SDAs formed and started to grow, (b) and (c) dendrite competition and primary dendrite arm spacing selection occurred, and (d) the dendrite tips reached the end of the domain; from this point until the end of the simulation run, secondary arm coarsening and final solidification of the liquid were achieved.

For each alloy, composition simulations were linked to a 1D heat transfer model allowing simulations to be run at a different position in the slab. In Figure 11, three such locations are used to illustrate this: (a) the chill surface, (b) 10 mm inside the slab, and (c) 50 mm inside the slab. The pertinent features are the coarsening of the microstructure; further into the slab, the simulation indicated that reducing heat transfer conditions led to a thicker solidified shell. Here, the primary difference was in the primary arm spacing increasing as the position moved into the cast slab. These simulations were further contrasted at a later point in the solidification progression in Figure 12 to highlight the impact on secondary arm spacing development in response to position in the slab ((a) chill surface, (b) 10 mm, and (c) 50 mm).

These results implied segregation into the liquid as solidification proceeds. As highlighted in the Thermo-Calc calculations, the segregation of elements is critical in issues such as nitrogen gas bubble formation. The results from a simulation of S32101 showing the segregation of each of the seven elements nominated in the project are shown in Figure 13. Micress provides the opportunity to probe how heat transfer conditions, microstructural development, and segregation interrelate. In the figure, the information is presented graphically with differences in concentration displayed by changing color levels. The underlying data at each of the 400,000 grid points is also accessible in Micress and can be manipulated using the Micress analysis software or extracted to conventional spreadsheets such as Excel for further analysis. In this way, the composition of the liquid phase during solidification can be plotted as a function of temperature (Figure 14).

Running Micress simulations enables to determine the effects of changing process parameters on the subsequent solidification. For example, by testing two superheating temperatures on the casting of S32101, we could extract relationships between fraction liquid as a function of temperature and the nitrogen segregation in the liquid (Figure 15). In this case, increasing the superheat led to a lower nitrogen composition in the liquid at a given temperature compared to superheating at 50 °C. As the nucleation of nitrogen gas bubbles is related to the content of dissolved nitrogen in the liquid, such an analysis offers opportunities for probing defect formation minimization.

The simulation domains comprised 400,000 grid points, and Micress outputted data on the position of the interface (in this case, solid and liquid). The average occupancy of grid points is directly related to the fineness of a given microstructure. Therefore, we can compare the influence of melt superheat on the resulting microstructure. In Figure 16, it can be observed that a higher superheat led to a coarser microstructure. Alternatively, the simulations can address heat transfer conditions such as heat flux Q at the slab surface. The base simulations were conducted with a fixed heat transfer coefficient of 0.25 W/cm^2^·K, in line with conventional casting practices. A comparative simulation for S32101 with a superheating temperature of 50 °C but with the heat transfer coefficient increased to 0.35 W/cm^2^·K was also run, and the simulation results at 10 mm in the slab are included in Figure 16. The critical point is that, through incorporating a 1D heat transfer model into the Micress simulations of solidification, cooling conditions in the caster can be adjusted consistently, allowing direct comparison to, for example, the measure of microstructure refinement that arises from increasing the heat transfer coefficient. However, all the other information entailed in the simulations, such as element segregation, secondary phase nucleation, and growth, can also be assessed simultaneously.

## 4. Conclusions

The concentric HTCM method was employed to study the heating and cooling stages of S32101 DSS. During the heating process, the GB presented γ → δ phase transformation. As the temperature increased, γ fully transformed to δ. The GBs enlarged and slipped before melting, and thermal etching appeared. These phenomena indicate that the steel would melt soon. The liquid phase preferably formed inside of or close to the δ GBs. During the solidification process, the cooling rate affected the microstructure of S32101 DSS; as the cooling rate increased, the solid’s size was finer, the morphology comprised smaller grains, and the solidification behavior in the body of the liquid pool commenced more quickly. The distance (D) of S/L IB motion decreased as the cooling rate increased. The size of the dendritic structure of δ decreased until it disappeared. The results demonstrate that concentric solidification offers a good platform for studying DSSs, forming a good testbed for subsequent simulations. The growth of precipitates was observed.

The experimental approach based on in situ HTCM was found to be helpful in studying the solidification process of S32101 DSS. Moreover, the data can be used as a benchmark for simulation. A simulation framework using Micress linked to a 1D heat transfer model that enables consistent analysis of solidification dynamics in duplex stainless steels across the whole cast slab was established. The analysis of phase transformations, solute segregation, and microstructural fineness, among other aspects, was demonstrated.

The establishment of the Micress framework to simulate the solidification process can dramatically address the limitations of experiments and reduce the experimental workload. Micress offers the opportunity to explore how heat transfer conditions, microstructural development, and separations relate. By revealing the elemental segregation of S32101 DSS under the cooling rates of 5 and 150 °C/min, Micress was proven to be a powerful tool to tackle the problems encountered in production.

## Figures and Tables

**Figure 1 materials-15-05517-f001:**
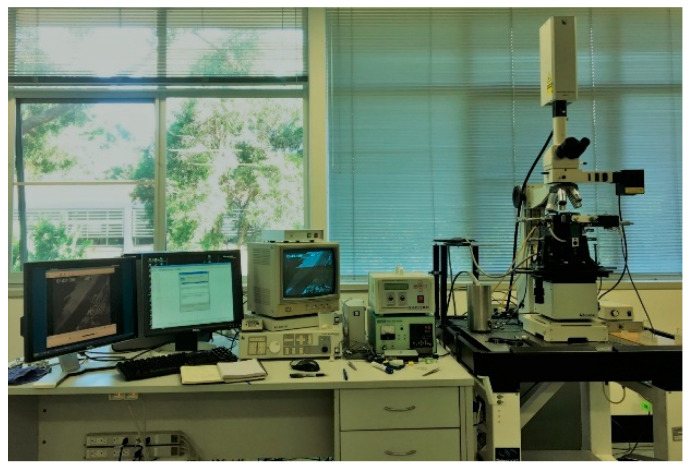
Picture of HTCM 1LM21H.

**Figure 2 materials-15-05517-f002:**
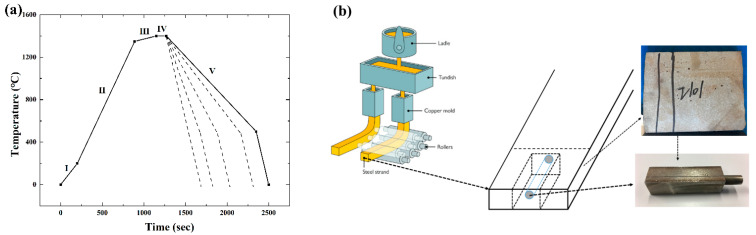
(**a**) Temperature profile of concentric HTCM experiments. I—Stabilizing stage, II—Fast heating stage, III—Slow heating stage, IV—Holding stage, and V—Cooling stage. Different dot lines represent different cooling rates; (**b**) schematic diagram and pictures of S32101 DSS samples.

**Figure 3 materials-15-05517-f003:**
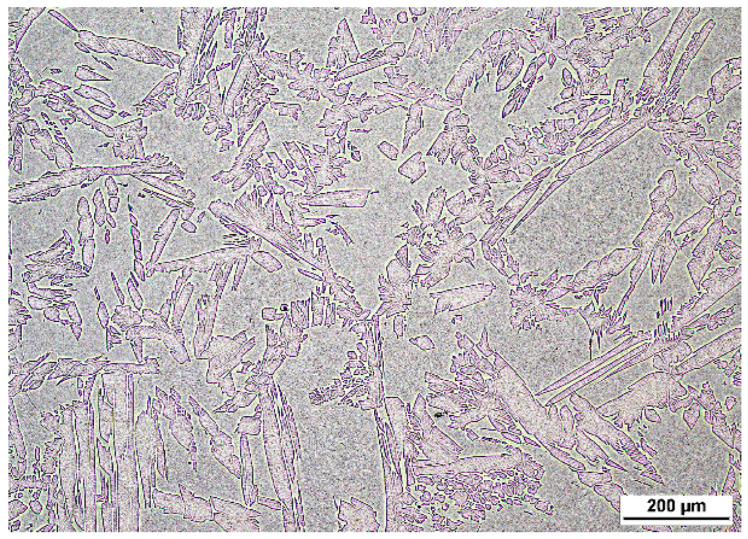
Optical microscopy of S32101 DSS cast with δ matrix and γ island structures.

**Figure 4 materials-15-05517-f004:**
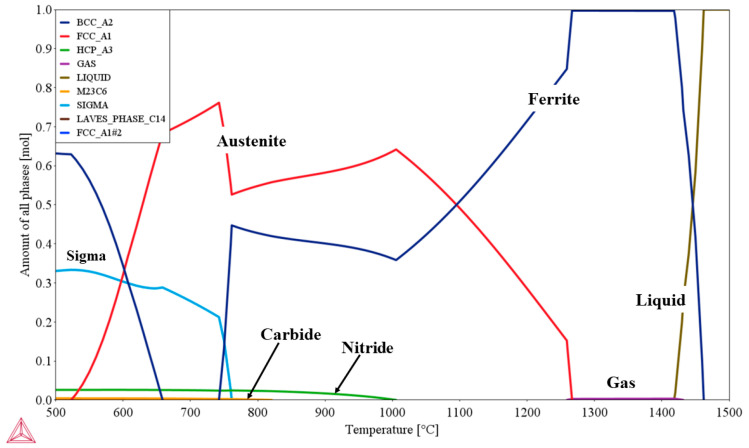
Property diagram of S32101 DSS calculated using Thermo-Calc showing the changes in all phases with temperature under equilibrium conditions.

**Figure 5 materials-15-05517-f005:**
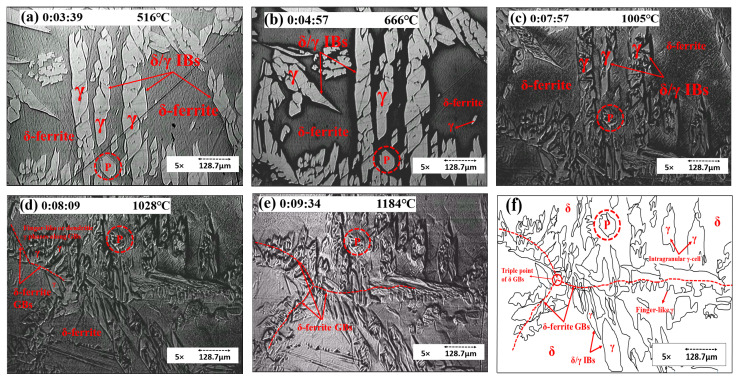
Concentric HTCM images revealing (**a**–**c**) the morphology changes of the γ phase and (**d**–**f**) the growth of δ/δ GBs; (**f**) a schematic diagram of the γ morphology of S32101 DSS.

**Figure 6 materials-15-05517-f006:**
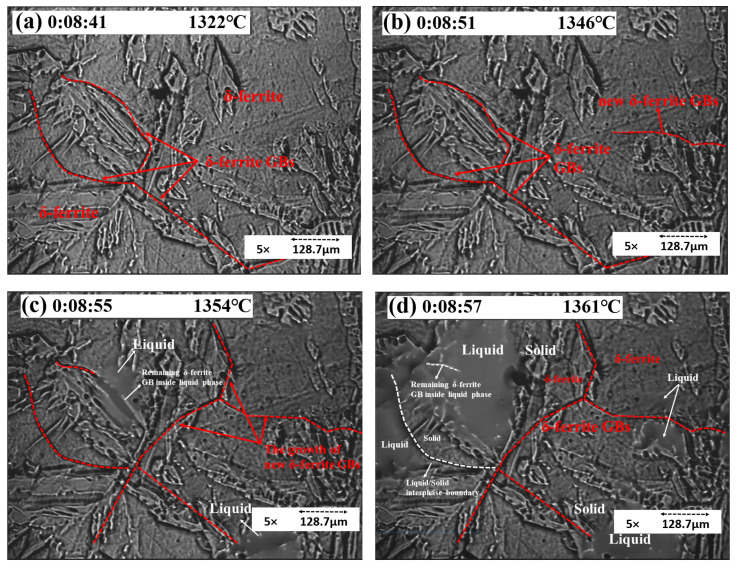
Concentric HTCM images revealing (**a**–**c**) the growth of δ GBs before melting and (**d**) the location of the liquid phase.

**Figure 7 materials-15-05517-f007:**
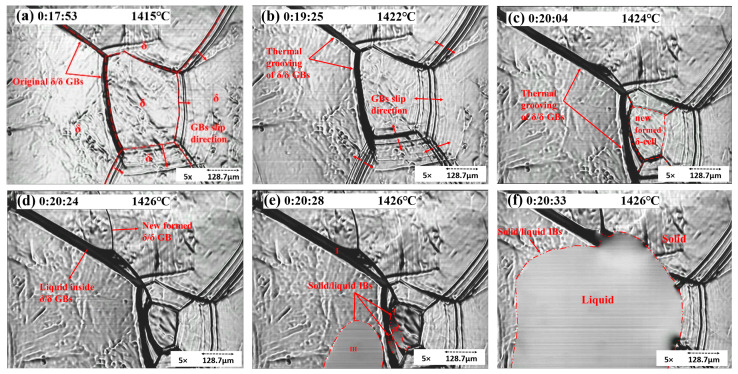
Concentric HTCM images revealing the δ/δ GBs enlarging and slipping (**a**–**c**) and S32101 DSS melting (**d**–**f**).

**Figure 8 materials-15-05517-f008:**
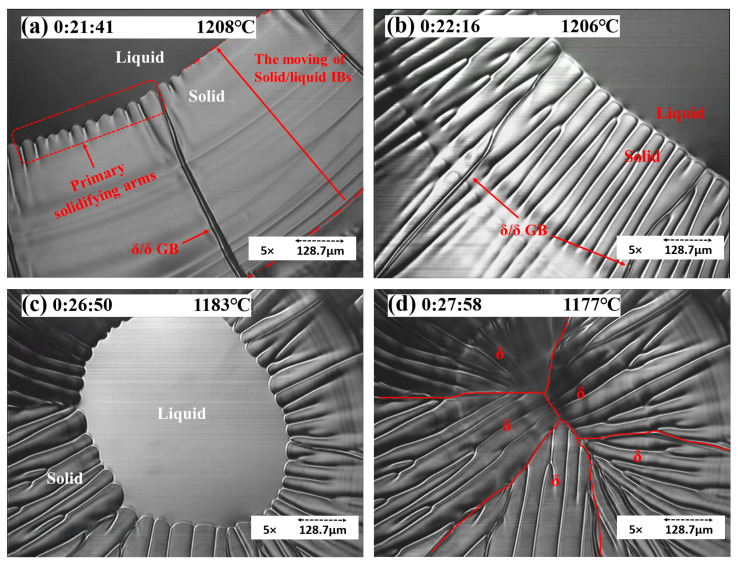
Concentric HTCM revealing the solidification process of S32101 under a 5 °C/min cooling rate: (**a**) the movement of S/L IB, (**b**) the growth of solidifying arms, (**c**) the residual liquid pool, and (**d**) the morphology of δ solids (cellular growth).

**Figure 9 materials-15-05517-f009:**
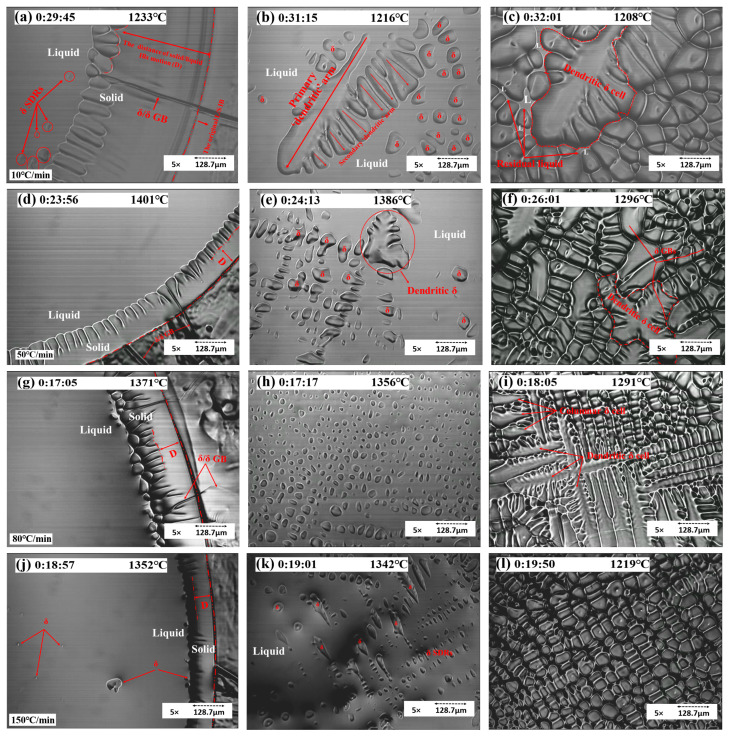
Concentric HTCM revealing the solidification process of S32101 under the cooling rates of (**a**–**c**) 10 °C/min (**d**–**f**) 50 °C/min (**g**–**i**) 80 °C/min, and (**j**–**l**) 150 °C/min.

**Figure 10 materials-15-05517-f010:**
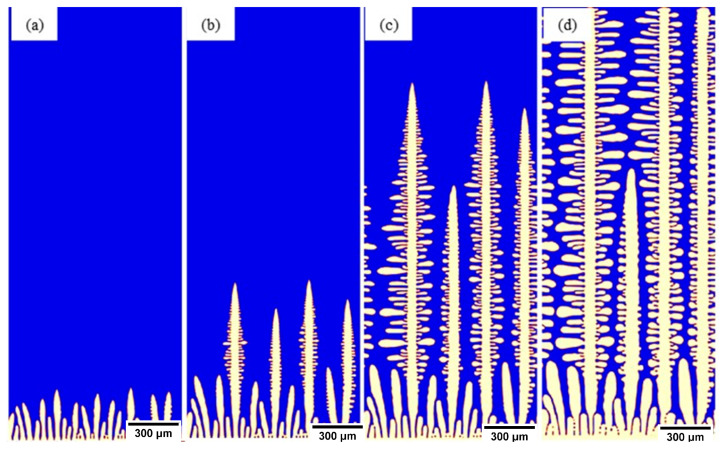
Micress simulation of S32101 chill surface with 100 °C superheat. (**a**) SDAs formation, (**b**) dendrite competition, (**c**) primary dendrite arm selection, and (**d**) dendrite tip across domain.

**Figure 11 materials-15-05517-f011:**
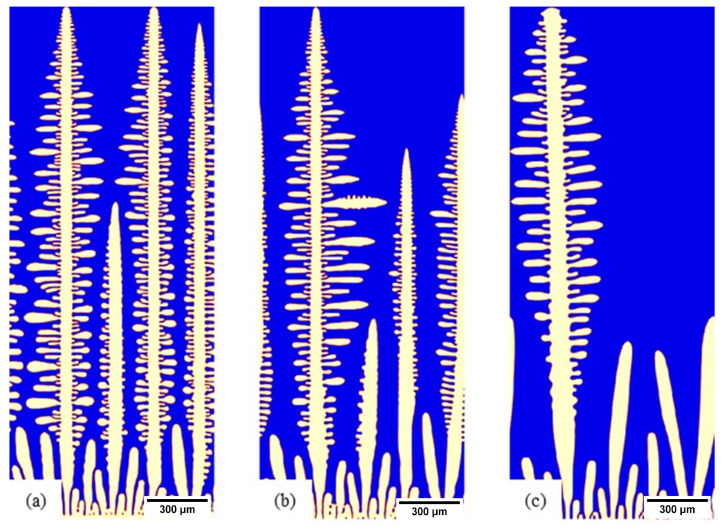
S32101, 100 °C superheated microstructure when growing dendrites reached the top of the simulation domain: (**a**) chill surface, (**b**) 10 mm in the slab, and (**c**) 50 mm in the slab.

**Figure 12 materials-15-05517-f012:**
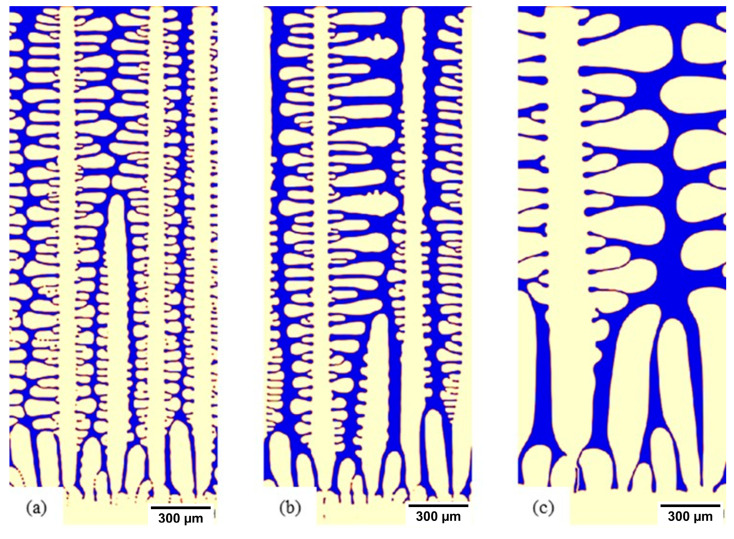
S32101, 100 °C superheated microstructure showing the development of SDAs: (**a**) chill surface, (**b**) 10 mm in the slab, and (**c**) 50 mm in the slab.

**Figure 13 materials-15-05517-f013:**
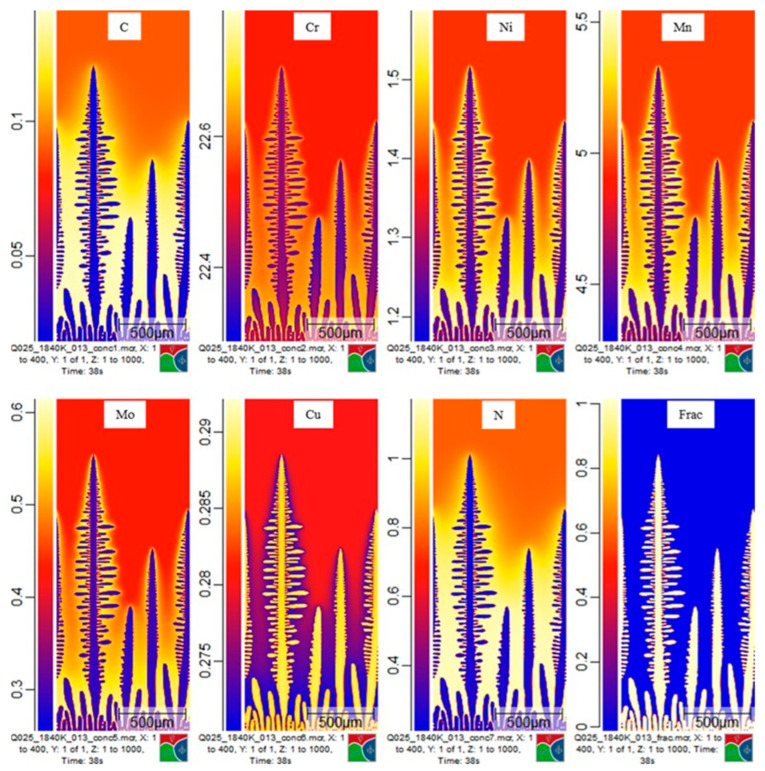
S32101, 100 °C superheated at 10 mm in the slab. Segregation of seven elements displayed from one timestep of a simulation run.

**Figure 14 materials-15-05517-f014:**
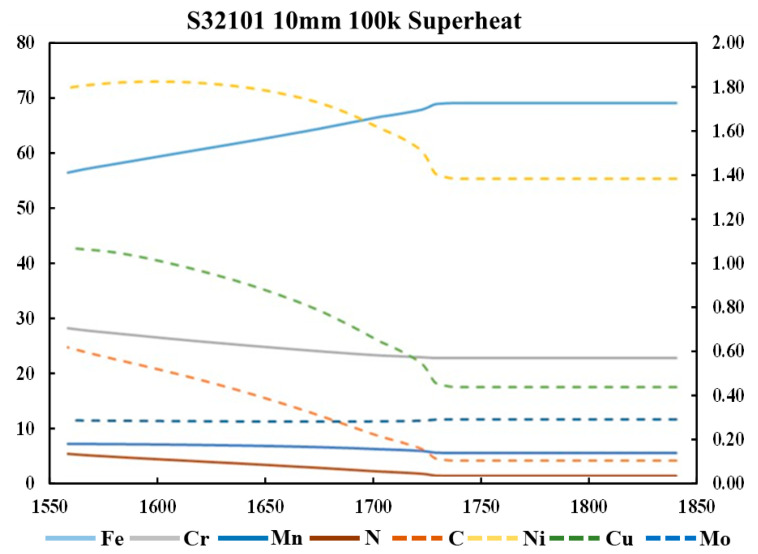
S32101, 100 °C superheated at 10 mm. The liquid phase’s average composition as a temperature function. Solid lines, left-hand axis; dashed lines, secondary axis.

**Figure 15 materials-15-05517-f015:**
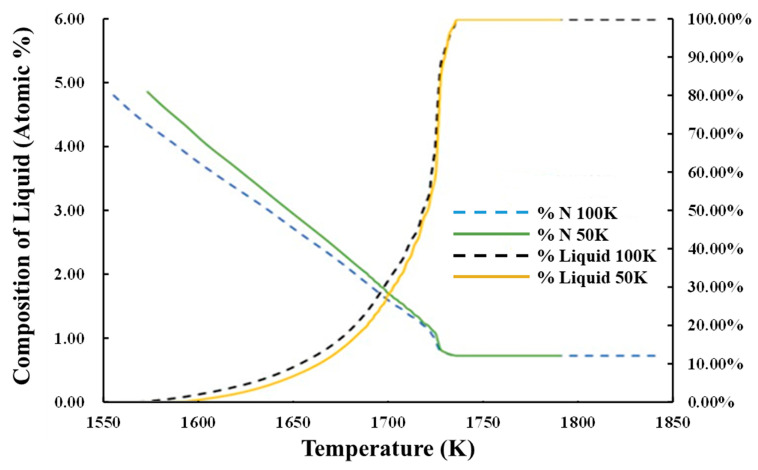
S32101 comparison of segregation of carbon and nitrogen in the liquid during solidification as a function of superheat (50 °C and 100 °C).

**Figure 16 materials-15-05517-f016:**
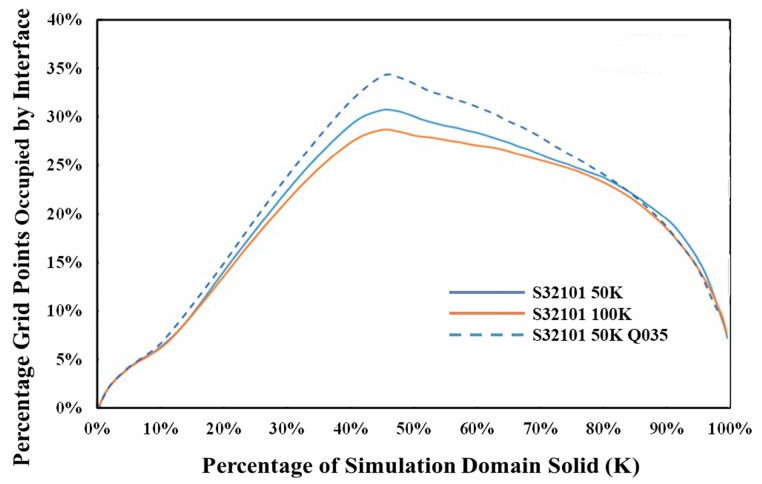
Influence of superheat on microstructural fineness of S32101, 50 °C and 100 °C superheat at 10 mm in the cast slab, when increasing heat flux at the boundary.

**Table 1 materials-15-05517-t001:** The chemical composition of S32101 (wt.%).

Grade	C	Mn	Cr	Ni	Mo	Cu	N
S32101	<0.04	5	21.5	1.5	0.5	0.5	0.2

**Table 2 materials-15-05517-t002:** Phases and phase-related parameters.

No.	Database ID	ID	Diffusion Data	µ_1/x_ (cm^4^/Js)	µ_2/x_ (cm^4^/Js)	σ_1/x_ (J/cm^2^)	σ_2/x_ (J/cm^2^)
1	FCC_A1	γ	MOBFE4		Diff Limit		0.8 × 10^−4^ to 1.2 × 10^−4^
2	BCC_A2	δ	MOBFE4	Diff Limit		0.8 × 10^−4^	
3	Liquid	L	Fixed	Diff Limit	1 × 10^−5^ to 1 × 10^−10^	0.8 × 10^−4^ to 1.2 × 10^−4^	0.3 ×10^−4^ to 1.2 × 10^−4^

## Data Availability

Data sharing is not applicable. No new data were created or analyzed in this study.

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
