# Peer review of "In Situ Observation and Phase-Field Simulation Framework of Duplex Stainless-Steel Slab during Solidification"

_materials, 2022, doi:10.3390/ma15165517_

Round 1

Reviewer 1 Report

Manuscript ID: materials-1840906

Title: In-situ Observation and Phase-field Simulation Framework of Duplex Stainless Steel Slab During Solidification

This manuscript is written decently. The manuscript deals with the solidifications as well as melting process in stainless steel. The confocal microscopy with temperature controller is functioned for the study of the microstructure. The Phase-Field Simulation: Micress was applied which might propose the reduction of experimental work load as well as limiting the production complications.

The accompanied consequences on the morphology and microstructure were revealed throughout this study by using the high temperature confocal microscope (HTCM). In general, the manuscript is acceptable. Just few minor comments:

-In the abstract section you should mention the most significant observation in the work . I mean the optimum acquired conditions.

-All abbreviation should be defined and to be consistent in the manuscript.

-Table 2 either to use E-04 or ×10-4

-The quality of Figure 2 better to be improved.

-Kindly revise the typing mistakes.

- Check guidelines once more to comply with all needed styles for the journal.

Best of luck

Reviewer 2 Report

The authors have done really good work which has scientific significance and can be used for industries. There are some minor queries which should be answered before the publication of this manuscript.

·       What is the basis of program set up of temperature? How these temperatures and time duration were chosen?

·       If possible, include the initial image and surface profile of Boasteel specifically in thickness direction.

·       Kindly give the proper citations for basic phase-field equation.

·       Figure 9 (h) is not explained clearly. What are these structures?

Overall the paper is good and can be accepted after these revisions.

Reviewer 3 Report

The authors of the manuscript made a very good study about the simulation of the solidification process during cooling for one stainless steel. The paper contains all of the necessary data and appropriate analysis of the topic to be accepted for publishing.

As the main contribution, in my opinion, we can point out the proof that the simulation can be used as a replacement for the experiment, based on this study. Of course, when it is possible the experiment should certainly be done but in order to reduce time and costs, the simulation can be a good solution. 

I also want to praise very good graphics with a pretty good denotation of the details. 

I only have a few remarks in order to improve the quality of the research:

- I suggest adding a little bit more details about the material used (such as steel application field, designation of the steel according to EN standard, ...) for example in section 2.1.2,

- Did you use a cooling medium during the cutting of the samples (Section 2.1.2)?

- Technical setting of the text, for example in row 23 you wrote the abbreviation DSS before the last abbreviated word, the are some extra spaces between words (row 203 after 500oC) as well as unnecessary spacing between paragraphs in the Conclusions.

Reviewer 4 Report

Dear authors,

Thank you for your exploration of how we can better understand cooling, solidification dynamics, and microstructure of duplex stainless steel. Here are some suggestions that should improve the quality of your manuscript.

1.     The hypothesis should be clearly written in Abstract as the aim, or a doubt, or a conclusion from the previous research that should be controlled /confirmed /observed in this study. Since this isn`t the first time observing the solidification dynamics, the hypothesis should have had clarified what were the authors planning to prove or what were they expecting to find as a scientific contribution in this research.

2.     In Introduction, line 44, the authors should explain the acronym.

3.     In Introduction, line 65, two references (if possible, from Web of Science) are needed for the Micress: where was it used, what has been proved, and why not another method.

4.     In Experimental and Modelling Approaches, line 76, the authors shouldn't keep ten references for proving something in only one sentence, and no proofs have been mentioned. That has to be separated in three or more groups of references with some comments in between.

5.     In Conclusion, the authors should write the findings, according to the hypothesis that was written in Abstract, and mentioned at the end of Introduction. Results of observation itself should have the purpose, and that should be the hypothesis in this manuscript.
